# Mental Health and Smoking-Related Determinants of Alcohol Drinking Experience in Korean Adolescents

**DOI:** 10.3390/ijerph20064724

**Published:** 2023-03-07

**Authors:** Sook Kyoung Park, Hae-Kyung Jo, Eunju Song

**Affiliations:** 1College of Nursing, Jeonbuk National University, Jeonju 54896, Republic of Korea; 2Department of Nursing, Jeon-ju University, Jeonju 55069, Republic of Korea; 3Department of Nursing, Wonkwang University, Iksan 54538, Republic of Korea

**Keywords:** adolescent, alcohol drinking, mental health, smoking

## Abstract

This study aimed to identify the determinants of mental health and smoking-related behaviors among Korean adolescents with drinking experience. Secondary data from the Korean Youth Risk Behavior Web-based Survey (2021) were analyzed. The final study sample consisted of data from 5905 adolescents with a history of smoking. Chi-square and multivariate logistic regression analyses were used to examine the factors related to drinking experience. The factors that influenced alcohol drinking were sex, school level, academic performance, self-reported depression, and smoking. The results of this study showed that there are numerous factors affecting adolescents’ drinking experience. Early education and interventions are needed to reduce alcohol consumption among adolescents. Integrated attention and support from society, school, and family are necessary so that they can cope with and adapt to stress in a healthy way.

## 1. Introduction

Drinking that begins in adolescence is associated with long-term effects, such as decreased memory and attention [1]. Neurotoxic effects are one of the many types of adverse effects of consuming alcohol at a developmentally vulnerable age, and this can cause changes in brain functions, increased susceptibility to diseases in adulthood, difficulties with emotional regulation, increased risky sexual behavior, fighting, and physical aggression [2]. If this drinking behavior continues during adolescence, it can lead to difficulties in academic achievement and negatively affect every life.

Owing to the 2019 Coronavirus Disease (COVID-19) pandemic, social restrictions and controls have been implemented in many countries. School closures and transitions to online classes have led to an increase in psychological difficulties among adolescents, such as depression and anxiety disorders, isolation, and interpersonal problems [3,4]. Adverse health behaviors have also been observed since the pandemic, such as smoking and alcohol use [5]. Subsequently, drinking alcohol is highly related to depression and behavioral problems, such as attention-deficit hyperactivity disorder (ADHD), which can occur when children or adolescents begin substance abuse early [6]. This also causes emotional problems, such as aggression, impulsivity, anxiety, and excitement, and is more likely to cause eating disorders than smoking [1,7]. Therefore, it is important to investigate adolescents’ mental health and substance use, such as smoking and drinking, during the pandemic.

Substance use is strongly associated with attempted and completed suicide [8,9]. Recently, the use of conventional cigarettes and e-cigarettes by adolescents has risen, and adolescents who drink frequently are more likely to use e-cigarettes [10], indicating an urgent need for preventive measures against substance abuse.

Many countries have conducted annual youth health behavior surveys among adolescents because of these risks. South Korea, specifically, has a higher prevalence of alcohol use than other countries and a culture of tolerance for drinking; greater access to alcohol incurs large social costs across the country [11,12]. Roughly half of all Korean adolescents have experienced alcohol consumption, and the prevalence of binge drinking among them has increased [10].

Accordingly, this study identifies the effects of general and mental health characteristics and smoking-related behaviors on adolescents’ drinking behaviors. This study utilized secondary data on youth health behaviors collected by the Korean government, with the aim of gaining insights into nursing interventions and alcohol abstinence education targeting adolescents. The specific research objectives were as follows:(1)Identify general characteristics, mental health characteristics, smoking behavior characteristics, and drinking experiences of adolescents.(2)Identify the factors that affect the drinking experience of adolescents.

## 2. Materials and Methods

### 2.1. Study Design

Since 2005, the Korea Centers for Disease Control and Prevention (KCDC) has been conducting the Korean Youth Risk Behavior Web-based Survey every year using a self-administered questionnaire to identify health-related behaviors, including alcohol use, smoking, and mental health status among first- to third-year high school students in Korea [13]. Secondary data from the 2021 survey were analyzed.

### 2.2. Study Population

The 17th Korean Youth Risk Behaviors Web-based Survey (2021) [13] data used in this study were divided into stratified sampling distributions and sampling stages of the population of middle and high school students nationwide as of April 2021, and then the sample was selected. A total of 54,848 students (92.9%) from 796 of 400 middle and 400 high schools participated in the survey. Of the participants, 5905 were smokers who answered “yes” to the question “Have you ever smoked?”.

In this data (a total of 54,848 students), 17,939 students answered “yes” to the question “Have you ever had more than one glass of alcohol?”.

### 2.3. Ethical Considerations

The n Youth Risk Behaviors Web-based Survey was conducted after review and approval by the Medical Research Ethics Review Committee from the KCDC regarding the scales and investigation process used for the ethical consideration of the participants. Prior consent was obtained from all participants before data collection. In addition, due to the secondary nature of the data, prior to the research, the researcher applied for an exemption from the Institutional Review Board of the institution to which the researcher belonged and was exempted from deliberation (JBNU 2022-06-014).

### 2.4. Data Analysis

Data analysis was performed using IBM SPSS Statistics for Windows v. 25.0 (IBM Corporation, Armonk, NY, USA). A chi-square test was conducted to assess the differences between the demographic, mental health, and smoking-related characteristics of Korean adolescents in the sample. Additionally, multivariate logistic regression analysis was performed using these variables. The weights for each variable in this study were already reflected in the original data provided by the KCDC, and the weights were calculated by multiplying the value obtained by multiplying the reciprocal of the extraction rate and the reciprocal of the response rate by the weight post-correction rate. Moreover, the calculation of biased results was controlled through the composite sample design, and the value was calculated using the weighted average.

#### 2.4.1. General Characteristics

Gender was indicated as “boy” or “girl”, school type was “middle school” or “high school”, and subjective academic performance over the previous 12 months was reclassified as “high”, “middle”, or “low.” Residence area was classified into “metropolitan cities”, “small cities”, or “rural area”, and the type of residence was reclassified into “living with family” or “living with someone other than family”. Economic status was reclassified as “high”, “middle”, and “low”.

#### 2.4.2. Mental Health Characteristics

Mental health characteristics included depression, stress perception, suicidal ideation, and perceived sleep satisfaction. Depression experience was classified as ‘yes’ or ‘no’ to the question “Have you ever felt so sad or hopeless that you stopped your daily life for 2 weeks in the past 12 months?” Perceived stress was assessed with the question “In general, how much stress do you usually feel?” Participants were asked to answer this question on a scale of “high”, “a little”, and “none”. Suicidal ideation was classified as “yes” or “no” to the question “Have you ever seriously thought of suicide in the past 12 months?” Perceived sleep satisfaction was reclassified into “sufficient”, “average”, and “insufficient” in response to “Do you think that you get adequate sleep to recover from fatigue on weekdays?”.

#### 2.4.3. Smoking-Related Characteristics

Characteristics of smoking behavior included the number of smoking days in the last 30 days and the time of first smoking. The number of smoking days in the last 30 days was classified as “non-smoking”, “1–2 days per month”, “3–5 days per month”, and “≥ 6 days per month”; the first smoking period was classified as “elementary school”, “middle school”, and “high school”.

## 3. Results

### 3.1. Characteristics of the Participants

The sociodemographic characteristics of this study were as follows: The gender of the sample showed a high proportion of boys, and most of the participants were high school students. In addition, 60.3% of students reported living in small cities or rural areas, and 46.6% of students reported belonging to middle-income households in terms of economic status.

Overall, 40.4% of all participants experienced depression, 45.9% of students reported they were highly aware of stress, and 21.5% of students reported that they had suicidal thoughts. More than half of the students had insufficient sleep. For smoking-related characteristics, only 28.8% of participants smoked only once. The time of smoking initiation was the highest in middle school.

Further details regarding the general characteristics of the study participants are shown in Table 1.

### 3.2. Differences in Drinking Experience According to the Characteristics of the Participants

Table 2 shows the chi-square results of the general characteristics of this sample.

For gender, it was found that there were more participants in the drinking experience group than those who had no drinking experience, which was statistically significant (t = 12.22, *p* < 0.001). In terms of school grade, there were more with drinking experience than those without drinking experience, which was statistically significant (t = 147.03, *p* < 0.001). In the case of academic achievement, those in the drinking experience group were higher than those in the group without drinking experience, and it was statistically significant (F^§^ = 4.83, *p* < 0.001).

There was a statistically significant difference in the area of residence between the drinking and non-drinking groups (t = 4.83, *p* = 0.028).

There was a statistically significant difference in economic status between the drinking and non-drinking groups (F^§^ = 5.85, *p* = 0.003).

### 3.3. The Effects of Adolescent Mental Health and Smoking Characteristics on Drinking Experience

A multiple logistic regression analysis was conducted to identify the effects of the general characteristics, mental health, and smoking-related characteristics of adolescents on their drinking experience (Table 3).

The following general characteristics were statistically significant: gender (boys) (OR = 1.98, CI: 1.77–2.16), middle school students (OR = 1.22, CI: 1.09–1.37), and low academic performance (OR = 2.18, CI: 1.94–2.45). Students with these characteristics showed an increased possibility of drinking. The *p* values were <0.001, <0.001, and 0.003, respectively.

Statistically significant factors in mental health characteristics were as follows. Students who responded that they were depressed (OR = 1.17, CI: 0.87–1.57) showed an increased possibility of drinking *(p* < 0.001). In contrast, students who had sufficient sleep (OR = 0.57, CI: 0.41–0.84) and students who slept an average amount (OR = 0.74, CI: 0.55–0.99) showed statistically significantly lower rates of drinking behavior. The *p*-values were 0.004 and 0.040, respectively. Smoking 1–2 times a month (OR = 1.48, CI: 1.04–2.11), 3–5 times a month (OR = 1.54, CI: 0.91–2.60), and over 6 times a month (OR = 1.421, CI: 1.04–1.84) significantly predicted an increased possibility of drinking. The *p*-values were 0.029, 0.004, and < 0.001, respectively.

In addition, starting smoking in middle school (OR = 2.55, CI: 1.73–3.75) and high school (OR = 2.67, CI: 1.65–4.31) significantly predicted drinking behavior (*p* <0.001).

## 4. Discussion

This study analyzed the factors influencing drinking experience based on data from the Korean Youth Risk Behavior Web-based Survey conducted by the KCDC in 2021. It was found that gender, school year, and academic performance were statistically significant predictors of drinking experience. Some studies have already reported that drinking behaviors were higher in male students [2,14]. This study found evidence consistent with this, possibly because female students had more positive attitudes than male students toward the early prevention of drinking or smoking [15]. However, in some recent studies, there has been little gender difference in drinking and smoking problems [16]. In 2020, blackout experiences of Korean adolescents due to binge drinking were higher among female than male students [17], and the prevalence of alcohol use among young Korean women was reportedly increasing [18]. Education on abstaining from alcohol consumption should be implemented without gender discrimination.

Results also indicated that the proportion of high school students who had experienced drinking was high, supporting a previous study showing that the risk of drinking increases with age [2]. However, in Korea, the age of first-time drinkers is decreasing [12]. The possibility of drinking in middle school students was found to be high, which suggests that all periods of adolescence are exposed to the risk of drinking. As such, it appears that strong state sanctions, such as the lockdown or social distancing during the pandemic, did not have a significant impact on the decline in alcohol consumption among students.

Furthermore, students with low grades drank more and were predicted to drink more. The highest stress among Korean adolescents was academic [19]. In Korea, which has a world-class educational zeal and college entrance rate, academic achievement can be a stressful factor at home and school. As this is a period of high interest in academic achievement, it is necessary to educate students about the effects of alcohol on the brain and its adverse effect on learning functions.

The influencing factor of residential area was not significant, but the drinking rate among students in small cities and rural areas was high. This was similar to the findings that drinking and binge drinking were higher among young people living in rural areas than in urban areas [20,21]. Students from middle and lower economic statuses also had a high drinking rate, which was linked to the high prevalence of alcohol use disorder among Korean adults in the low-income group [18]. In contrast, some studies have reported that alcohol and tobacco consumption rank high among public health priorities for young adolescents in affluent countries [14]. The Korean economy is the world’s 10th largest in terms of scale and 18th largest in per capita income among the Organization for Economic Co-operation and Development (OECD) countries [22]. However, in this study, economic status was not a statistically significant factor influencing adolescents’ drinking behaviors.

Among the mental health characteristics, depression was statistically significant in drinking experience and possibility, but suicidal ideation was not. Despite the results of this study, suicide attempts are higher when substance abuse begins at an early age [23]. In South Korea, suicide was reported as the number one cause of death among adolescents in 2021, during the pandemic, and suicide has been the number one cause of death among adolescents in the last nine years [24]. Therefore, early detection of depression can be achieved by assessing mental health during drinking interventions in adolescents. This will help prevent drinking, which is a risk factor for suicide, and provide adolescents with the tools to positively cope with risks to their mental well-being.

Among the mental health characteristics, students who did not get enough sleep had a significantly higher drinking experience rate and a higher possibility of drinking experience. Insomnia rates among Korean adolescents are extremely high, mainly due to excessive academic pressure [25]. Sleep problems can continue to occur during adolescence when trying to keep up with schoolwork. Short sleep times and sleep dissatisfaction in adolescents are also associated with hypertension, obesity, depression, and suicidal ideation [26]. It has been reported that the main cause of relapse in patients with alcohol use disorder is the use of alcohol as a sleep inducer [27,28]. This suggests that it is necessary to closely reflect the relationship between sleep and alcohol in educational content from scientific and intellectual points of view.

The longer the number of days smoked, the higher the rate of drinking experience, which became an influencing factor. In the pre-COVID-19 surveys, high school students had a high smoking rate [29,30], consistent with this study’s findings. In Korea, smoking prevention and anti-smoking projects for teenagers started in 1999 to lower the youth smoking rate, and in 2015, they expanded to schools nationwide, while the school smoking prevention project was being implemented for all students [31]. Nevertheless, in this study, there were cases in which elementary school students also smoked, and it is necessary to pay attention to this because the factors influencing drinking among middle school students who smoked were significant. In addition, the results of this study are related to the fact that the possibility of drinking among middle school students is higher than that among high school students due to “eighth-grade syndrome”, a South Korean term indicating the negative tendency of adolescents to be rebellious and aggressive, which is the crux of a cultural joke that North Korea cannot invade South Korea because of eighth-grade syndrome [32]. The present findings can be considered an indicator of the urgent need for preventive education for lower grades.

## 5. Conclusions

Recently, alcohol consumption and smoking rates among adolescents have declined globally [1,14]. South Korea also showed a similar trend, with drinking and smoking rates in 2020 and 2021 decreasing by 3–4% compared to 2019 [13]. However, this trend in adolescents was interpreted as the effect of online classes and social distancing due to the outbreak of COVID-19, rather than improved health-related behavior. In South Korea, less than 20% of drinking prevention education is conducted in schools [33]; thus, it is insufficient compared to mandatory smoking education. In addition, the pandemic prevented these programs from being actively conducted. This study found that strong control measures, such as social distancing cannot control substance use. It is impossible to predict how freedom from the endemic era will affect substance use among adolescents. However, integrated attention and support from society, school, and family are necessary for adolescents to cope with and adapt to stress in a healthy way.

## Figures and Tables

**Table 1 ijerph-20-04724-t001:** General characteristics of participants (N = 5905).

Variables	Categories	n ^†^	% ^‡^
Gender	Boys	4008	68.9
	Girls	1897	31.1
School grade	Middle school	1702	25.0
	High school	4203	75.0
Academic performance	High	1512	25.6
	Middle	1560	26.3
	Low	2833	48.1
Area of residence	Metropolitan	2400	39.7
	Small city or rural area	3505	60.3
Type of living	With family	5468	93.2
	Outside of family.	437	6.8
Economic status	High	2109	36.7
	Middle	2771	46.6
	Low	1025	16.7
Experience of depression	Yes	2394	40.4
	No	3511	59.6
Perceived stress level	High	2733	45.9
	Little	2175	36.9
	None	997	17.2
Suicidal ideation	Yes	1277	21.5
	No	4628	78.5
Perceived sleep satisfaction	Sufficient	1011	17.5
	Average	1710	28.7
	Insufficient	3184	53.8
Smoking frequency(the last month)	No	1139	28.8
	1~2 days/month	772	19.4
	3~5 days/month	369	9.6
	≥6 days/month	1595	42.1
The first day of smoking	Elementary school	295	7.4
	Middle school	2585	64.6
	High school	1033	28.0

^†^ Unweighted count; ^‡^ weighted %.

**Table 2 ijerph-20-04724-t002:** Differences in drinking experience according to characteristics (N = 5905).

Variables	Categories	Experience of Drinking				F^§^ or t (*p*)
		Yes (4997)		No (908)		
		n^†^	%^‡^	n^†^	%^‡^	
Gender	Boys	3333	83.9	675	16.1	12.22 (<0.001)
	Girls	1664	87.8	233	12.2	
School grade	Middle school	1272	74.2	430	25.8	147.03 (<0.001)
	High school	3725	88.7	478	11.3	
Academic performance	High	1205	80.0	307	20.0	19.31 (<0.001)
	Middle	1334	86.3	226	13.7	
	Low	2458	87.1	375	12.9	
Area of residence	Metropolitan city	1988	83.6	412	16.4	4.83 (0.028)
	Small city or rural areas	3009	86.0	496	14.0	
Type of living	With family	4635	85.1	833	14.9	0.26 (0.612)
	Outside of family	362	84.1	75	15.9	
Economic status	High	1746	83.3	363	16.7	5.85 (0.003)
	Middle	2349	85.3	422	14.7	
	Low	902	88.5	123	11.5	
Experience of depression	Yes	2095	87.8	299	12.2	18.77 (<0.001)
	No	2902	83.2	609	16.8	
Perceived stress level	High	2343	86.2	390	13.8	3.64 (0.027)
	Middle	1843	85.0	332	15.0	
	Low	811	82.2	186	17.8	
Suicidal ideation	Yes	1095	86.2	182	13.8	1.33 (0.249)
	No	3902	84.8	726	15.2	
Perceived sleep satisfaction	Sufficient	797	80.9	214	19.1	13.44 (<0.001)
	Average	1424	83.4	286	16.6	
	Insufficient	2776	87.3	408	12.7	
Smoking frequency(the previous month)	No	978	86.8	161	13.2	9.24 (<0.001)
	1~2 days/month	685	89.1	87	10.9	
	3~5 days/month	331	89.9	38	10.1	
	≥6 days/month	1482	93.2	113	6.8	
The first day of smoking	Elementary school	226	77.7	69	22.3	25.75 (<0.001)
	Middle school	2287	89.0	298	11.0	
	High school	960	93.0	73	7.0	

^†^ Unweighted count; ^‡^ weighted %; ^§^ Rao–Scott composite sample chi-square test.

**Table 3 ijerph-20-04724-t003:** Factors of mental health and smoking-related characteristics on drinking experience in adolescents.

Variables	Categories	Odds Ratio	95% Confidence Interval	*p*
(constant)		8.49	4.22–17.07	<0.001
Gender	Boys	1.98	1.77–2.61	<0.001
	Girls	Ref		
School grade	Middle school	1.54	0.83–1.68	<0.001
	High school	Ref		
Academic performance	High	Ref		
	Middle	1.08	0.77–1.51	0.646
	low	2.18	1.94–2.45	0.003
Area of residence	Small city or rural area	Ref.		
	Metropolitan city	0.76	0.57–0.99	0.47
Economic status	High	0.80	0.53–1.20	
	Middle	0.76	0.51–1.14	0.268
	Low	Ref.	0.90–1.23	0.188
Experience of depression	Yes	1.17	0.87–1.57	<0.001
	No	Ref		
Perceived stress level	High	0.96	0.64–1.43	0.829
	Middle	1.03	0.70–1.53	0.870
	Low	Ref		
Perceived sleep satisfaction	Sufficient	0.57	0.41–0.84	0.004
	Average	0.74	0.55–0.99	0.040
	Insufficient	Ref		
Smoking frequency(the last month)	No	Ref.		
	1~2 days/month	1.48	1.04–2.11	0.029
	3~5 days/month	1.54	0.91–2.60	0.004
	≥6 days/month	1.42	1.04–1.84	<0.001
The first day of smoking	Elementary school	Ref.		
	Middle school	2.55	1.73–3.75	<0.001
	High school	2.67	1.65–4.31	<0.001

Pseudo R2 = 0.78, Wald F = 8.35, *p* < 0.001.

## Data Availability

Publicly available datasets were analyzed in this study. This data can be found here: https://www.kdca.go.kr/yhs/ accessed on 29 April 2022.

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
