# Peer review of "Mental Health and Smoking-Related Determinants of Alcohol Drinking Experience in Korean Adolescents"

_ijerph, 2023, doi:10.3390/ijerph20064724_

Round 1
Reviewer 1 Report
Thank you for the opportunity to review your manuscript, which reports a predictive study based on secondary data from the Korean Youth Risk Behavior Web-based Survey (2021). The matter of health-risk behaviours is important to address due to the various long-term effects on individuals, on societal resources, and on public health systems. The study findings confirm a need to focus on interventions targeted towards specific demographics
Other than my recommendation for a major revision of the results section, my suggestions are for minor corrections here and there throughout the manuscript. I have inserted highlights within the manuscript either to indicate where attention needs to be directed to wording, or to flag some specific comments or queries for you to take into consideration. These are detailed below.
Abstract
Lines 5-6. “Secondary data from the Korean Youth Risk Behavior Web-based Survey (2021) was analyzed.” Data is the plural form, so data were analyzed is appropriate.
Lines 8-10. “The factors that influenced alcohol drinking were boys, middle school students, low level of academic performance, students who responded that they were depressed, and students who smoked a lot were more likely to drink alcohol.” Here you specify factors but then refer to a particular level within each factor (i.e., sex, school level, academic performance level, self-reported depression, smoking). Being a boy, etcetera, isn’t itself a factor per se.
Line 11. “These factors were significantly predicted an increased possibility of alcohol drinking.” Delete were.
Introduction.
Line 20. “Neurotoxic effects are one of the many adverse effects…” Should refer to ‘one of the many types of adverse effects’.
Line 25. “…and negatively affect real-life.” There is no need for the hyphen: real life. However, I’m not sure how you’re differentiating ‘real life’ here from other forms of existence. Do you perhaps intend to refer to everyday life?
Lines 55-56. “Identify the factors influencing the drinking experience of general characteristics, mental health characteristics, and smoking behavior characteristics of adolescents.” The wording of your second research objective is not clear with respect to meaning. For instance, what is the referent for ‘the drinking experience of X, Y and Z characteristics’?
Materials and Method
Lines 62-63. Data were analyzed.
Line 66. “divided into stratified, sampling distribution, and sampling stages” Stratified what? If a stratified sampling distribution, no need for the comma.
2.2 Study population. The first line of your abstract indicates your sample all had drinking experience: “This study aimed to identify the determinants of mental health and smoking-related behaviors of Korean adolescents with drinking experience.” In section 2.2, you indicate how many participants had smoked; why not also report here how many had drinking experience?
Line 92. You refer to ‘subjective academic performance’; does that mean participants were asked to self-report how they were performing academically? Should be made clear.
Line 101. “‘Do you feel it?' were reclassified as ‘yes’ and ‘no.” Unclear what ‘Do you feel it’ refers to, and this is a singular item so should be ‘was reclassified’. Also need a closing quotation mark for ‘no’.
Line 106. Questions should be question because just the one question stated here.
Lines 120-122. References to ‘people’ should be ‘students’ for consistency and to be clear still referring to the sample.
Results
Section 3.1 Characteristics of the participants. Avoid repeating in text any content reported in a table. This section can be abbreviated by referring to counts and percentages in Table 1.
Section 3.2 Participants’ drinking experience. This section could be deleted if this statement is placed in section 2.2 where the percentage with smoking experience is reported. Even if you choose to retain this section here, Table 2 is redundant because repeats what’s stated in text.
Section 3.3. This entire section needs rewording for clarity and adequacy of reporting. What type of statistical test/s did you perform?
“In the group with no drinking experience by gender, 675 boys (16.1%), 233 girls (12.2%), 3,333 boys (83.9%), and 1,664 girls (87.8%) were statistically significant (p <.001).” You need to explain what difference you were testing for here that was found to be statistically significant. And if all had no drinking experience, what do the different counts and percentages represent for boys and girls?
“In terms of school type, 430 middle school students (25.8%) and 478 high school students (11.3%) were found to have no drinking experience, while 1,272 middle school students (74.2%) and 3,725 high school students (88.7%) were statistically significant (p <.001).” Here you’re saying that some students had no drinking experience and some were statistically significant, but you haven’t explained what difference you tested for, or what the different counts and percentages represent.
“In the case of academic achievement over the past 12 months, 307 students (20.0 %) in the group without drinking experience responded with high grade, 226 middle grade (13.7%), and 375 students (12.9%); the drinking experience group was statistically significant (p 151 <.001) with the highest 1,205 (80.0%), the middle 1,334 (86.3%), and the lower 2,458 152 (87.1%).” What does it mean for the ‘drinking experience group’ to be statistically significant? On what point of difference? Further, you haven’t completed the phrase to explain what the 375 students refers to.
For the paragraph in lines 154-164, you need to specify the nature of each significant difference (i.e., in residence area, in economic status). For example, are you reporting that a statistically larger portion of the sample with drinking experience reside in a particular category of residence area? If so, make clear what this is, and the same for economic status. Reporting is not clear at all currently.
Lines 155-156. “In terms of residence, 412 (16.4%) lived in the non-drinking experience group in metropolitan cities…” Here you’re saying some students lived in a group as opposed to a location. Needs rewording for clarity.
Line 163. You refer to the availability of ‘further details’ in Table 3, but it appears that Table 3 content replicates what’s reported in text. The substantial rewriting of section 3.3 would benefit from clear and early reference to Table 3 and avoiding any repetition of content within text. Reporting could then focus on appropriate and adequate explanation of the statistical analyses and outcomes.
Section 3.4. Again, avoid repeating context in text when summarised in a Table.
Lines 179-180. “showed a statistically significant tendency to decrease drinking.” Well, unless you have data to suggest that these students were drinking a lot and subsequently drank less, you’re not reporting a decrease in drinking but rather a lower likelihood of them engaging in drinking behaviour or lower rates of reported drinking experiences compared to those in the other group/s.
Discussion
Lines 191-192. “This study analyzed the factors influencing drinking experience based on the Korea Youth Risk Behavior Web-based Survey data conducted by the KCDC in 2021.” The wording needs revision here; the survey was conducted, not the data. A suggested revision: This study analyzed the factors influencing drinking experience based on data from the Korea Youth Risk Behavior Web-based Survey conducted by the KCDC in 2021.
Line 206. “The possibility of drinking in middle school students was found to be high among, suggesting …”. Sentence incomplete – high among whom?

Author Response
We thank you and the reviewers for appreciating our work. We have revised the manuscript per the comments and suggestions of the reviewers. We hope that the revised manuscript is suitable for publication in International Journal of Environmental Research and Public Health.
We have revised the manuscript according to the suggestions and comments. We have uploaded the Marked-up Manuscript (manuscript with revisions highlighted), Manuscript (revised main text w/o track changes). Our responses to the comments are provided below.
Abstract
Lines 5-6. “Secondary data from the Korean Youth Risk Behavior Web-based Survey (2021) was analyzed.” Data is the plural form, so data were analyzed is appropriate.
- Response: We thank you for your comment. We apologize for this. We have changed ‘was’ to ‘were’ in the main text.
Lines 8-10. “The factors that influenced alcohol drinking were boys, middle school students, low level of academic performance, students who responded that they were depressed, and students who smoked a lot were more likely to drink alcohol.” Here you specify factors but then refer to a particular level within each factor (i.e., sex, school level, academic performance level, self-reported depression, smoking). Being a boy, etcetera, isn’t itself a factor per se.
- Response: We thank you for your comment. We have changed ‘sex, school level, academic performance level, self-reported depression, and smoking’ in the main text.
Line 11. “These factors were significantly predicted an increased possibility of alcohol drinking.” Delete were.
- Response: We appreciate your positive comment. We deleted the sentence.
Introduction.
Line 20. “Neurotoxic effects are one of the many adverse effects…” Should refer to ‘one of the many types of adverse effects’.
- Response: We agree with the comment. We revised it as you recommended.
Line 25. “…and negatively affect real-life.” There is no need for the hyphen: real life. However, I’m not sure how you’re differentiating ‘real life’ here from other forms of existence. Do you perhaps intend to refer to everyday life?
- Response: We appreciate your positive comment. We deleted the hyphen.
Lines 55-56. “Identify the factors influencing the drinking experience of general characteristics, mental health characteristics, and smoking behavior characteristics of adolescents.” The wording of your second research objective is not clear with respect to meaning. For instance, what is the referent for ‘the drinking experience of X, Y and Z characteristics’?
- Response: We thank you for the valuable comment. We have revised it in the main text.
“ Identify the factors that affect the drinking experience of adolescents.”
Materials and Method
Lines 62-63. Data were analyzed.
- Response: We thank you for this comment. We have corrected this error in the main text.
Line 66. “divided into stratified, sampling distribution, and sampling stages” Stratified what? If a stratified sampling distribution, no need for the comma.
- Response: We agree with the comment. We deleted the comma.
2.2 Study population. The first line of your abstract indicates your sample all had drinking experience: “This study aimed to identify the determinants of mental health and smoking-related behaviors of Korean adolescents with drinking experience.” In section 2.2, you indicate how many participants had smoked; why not also report here how many had drinking experience?
- Response: We agree with the comment. We added that sentence about Korean adolescents with drinking experience in the data.
“Incidentally, in this data (total of 54,848 students), 17,939 students answered “yes” to the question, “Have you ever had more than one glass of alcohol?”
Line 92. You refer to ‘subjective academic performance’; does that mean participants were asked to self-report how they were performing academically? Should be made clear.
- Response: We agree with the comment. We revised it as you recommended.
Line 101. “‘Do you feel it?' were reclassified as ‘yes’ and ‘no.” Unclear what ‘Do you feel it’ refers to, and this is a singular item so should be ‘was reclassified’. Also need a closing quotation mark for ‘no’.
- Response: We agree with the comment. We deleted and revised it as you recommended.
Line 106. Questions should be question because just the one question stated here.
- Response: We agree with the comment. We revised it as you recommended.
Lines 120-122. References to ‘people’ should be ‘students’ for consistency and to be clear still referring to the sample.
- Response: We thank you for your comment. We apologize for this error. We have changed ‘people’ to ‘students’ in the main text.
Results
Section 3.1 Characteristics of the participants. Avoid repeating in text any content reported in a table. This section can be abbreviated by referring to counts and percentages in Table 1.
- Response: We thank you for your comment. We have rewritten and revised about Table 1.
Section 3.2 Participants’ drinking experience. This section could be deleted if this statement is placed in section 2.2 where the percentage with smoking experience is reported. Even if you choose to retain this section here, Table 2 is redundant because repeats what’s stated in text.
- Response: We agree with the comment. We deleted the Table 2 and sentence.
Section 3.3. This entire section needs rewording for clarity and adequacy of reporting. What type of statistical test/s did you perform?
- Response: We thank you for your comment. We have rewritten and revised
“In the group with no drinking experience by gender, 675 boys (16.1%), 233 girls (12.2%), 3,333 boys (83.9%), and 1,664 girls (87.8%) were statistically significant (p <.001).” You need to explain what difference you were testing for here that was found to be statistically significant. And if all had no drinking experience, what do the different counts and percentages represent for boys and girls?
- Response: We thank you for your comment. We have rewritten and revised
“In terms of school type, 430 middle school students (25.8%) and 478 high school students (11.3%) were found to have no drinking experience, while 1,272 middle school students (74.2%) and 3,725 high school students (88.7%) were statistically significant (p <.001).” Here you’re saying that some students had no drinking experience and some were statistically significant, but you haven’t explained what difference you tested for, or what the different counts and percentages represent.
- Response: We thank you for your comment. We apologize for this. We have changed and revised.
“In the case of academic achievement over the past 12 months, 307 students (20.0 %) in the group without drinking experience responded with high grade, 226 middle grade (13.7%), and 375 students (12.9%); the drinking experience group was statistically significant (p 151 <.001) with the highest 1,205 (80.0%), the middle 1,334 (86.3%), and the lower 2,458 152 (87.1%).” What does it mean for the ‘drinking experience group’ to be statistically significant? On what point of difference? Further, you haven’t completed the phrase to explain what the 375 students refers to.
- Response: We thank you for your comment. We have changed and revised.
For the paragraph in lines 154-164, you need to specify the nature of each significant difference (i.e., in residence area, in economic status). For example, are you reporting that a statistically larger portion of the sample with drinking experience reside in a particular category of residence area? If so, make clear what this is, and the same for economic status. Reporting is not clear at all currently.
- Response: We thank you for the valuable comment. We have changed and revised it.
Lines 155-156. “In terms of residence, 412 (16.4%) lived in the non-drinking experience group in metropolitan cities…” Here you’re saying some students lived in a group as opposed to a location. Needs rewording for clarity.
- Response: We agree with the comment. We revised it as you recommended.
Line 163. You refer to the availability of ‘further details’ in Table 3, but it appears that Table 3 content replicates what’s reported in text. The substantial rewriting of section 3.3 would benefit from clear and early reference to Table 3 and avoiding any repetition of content within text. Reporting could then focus on appropriate and adequate explanation of the statistical analyses and outcomes.
- Response: We agree with the comment. We deleted ‘further detail’ and revised it.
Section 3.4. Again, avoid repeating context in text when summarised in a Table.
- Response: We agree with the comment. We deleted avoiding repeating context and revised it.
Lines 179-180. “showed a statistically significant tendency to decrease drinking.” Well, unless you have data to suggest that these students were drinking a lot and subsequently drank less, you’re not reporting a decrease in drinking but rather a lower likelihood of them engaging in drinking behaviour or lower rates of reported drinking experiences compared to those in the other group/s.
- Response: We appreciate your positive comment. We have rewritten and revised
Discussion
Lines 191-192. “This study analyzed the factors influencing drinking experience based on the Korea Youth Risk Behavior Web-based Survey data conducted by the KCDC in 2021.” The wording needs revision here; the survey was conducted, not the data. A suggested revision: This study analyzed the factors influencing drinking experience based on data from the Korea Youth Risk Behavior Web-based Survey conducted by the KCDC in 2021.
- Response: We agree with the comment. We revised it as you recommended.
Line 206. “The possibility of drinking in middle school students was found to be high among, suggesting …”. Sentence incomplete – high among who
- Response: We thank you for the comment. We revised it.
“The possibility of drinking in middle school students was found to be high which suggests that all periods of adolescence are exposed to the risk of drinking.”
To correct the grammar and paragraphs, the revised manuscript has once again been checked by a professional editor (“Editage” company in South Korea) to improve the technical and linguistic completeness of the manuscript.
We have carefully considered the reviewers’ comments and have made the necessary revisions in the manuscript. We hope that the revised manuscript is suitable for publication in International Journal of Environmental Research and Public Health.
Reviewer 2 Report
Report on “Mental Health……Korean Adolescents” (ID: IJERPH-2235904) for International Journal of Environmental Research and Public Health
This paper presents a statistical analysis of mental health with smoking and alcoholism among Korean adolescents.
The authors overall have done a good job. I find the analysis well-done and therefore I do think that the study should be published.
The econometric (statistical) analysis is fairly simple and straight-forward (which not necessarily a criticism of this work).
However, I think that the results and findings should be presented in a more professional and at the same time more reader-friendly way. I would suggest the results from the tables 3 and 4 in subsections 3.3 and 3.4 must be first presented properly, perhaps as a bullet-point lists and then discussed with possible policy implications.
The study otherwise is clear. The authors should do a final editorial check for any possible typos or grammatical mistakes, if any.
To sum up, I do think this paper has discussed something nice and thus should be accepted for publication; however, the authors should present and discuss all the findings properly.
Author Response
We thank you and the reviewers for appreciating our work. We have revised the manuscript per the comments and suggestions of the reviewers. We hope that the revised manuscript is suitable for publication in International Journal of Environmental Research and Public Health.
We have revised the manuscript according to the suggestions and comments. We have uploaded the Marked-up Manuscript (manuscript with revisions highlighted), Manuscript (revised main text w/o track changes). Our responses to the comments are provided below.
This paper presents a statistical analysis of mental health with smoking and alcoholism among Korean adolescents.
The authors overall have done a good job. I find the analysis well-done and therefore I do think that the study should be published.
The econometric (statistical) analysis is fairly simple and straight-forward (which not necessarily a criticism of this work).
However, I think that the results and findings should be presented in a more professional and at the same time more reader-friendly way. I would suggest the results from the tables 3 and 4 in subsections 3.3 and 3.4 must be first presented properly, perhaps as a bullet-point lists and then discussed with possible policy implications.
Response: We thank you for the comment. We have modified both the descriptions in 3.3 and 3.4 from table 3, 4. And then, We have revised most of the resulting statements.
The study otherwise is clear. The authors should do a final editorial check for any possible typos or grammatical mistakes, if any.
Response: We have absolutely agreed your comments. For your comments, first, we read each line of the manuscript and modified it to suit the context. Next, to correct the grammar and paragraphs, the revised manuscript has once again been checked by a professional editor (“Editage” company in South Korea) to improve the technical and linguistic completeness of the manuscript.
We have carefully considered the reviewers’ comments and have made the necessary revisions in the manuscript. We hope that the revised manuscript is suitable for publication in International Journal of Environmental Research and Public Health.

Round 2
Reviewer 1 Report
Thank you for the opportunity to review your revised paper, which is improved with respect to written expression. A number of corrections are still required.
My remaining concern is the copious repetition in text of content reported in tables. This must be avoided and I have provided some suggestions for revision. Additionally, conventional standards should be followed for reporting of statistical tests; again, I have provided an example of a possible revision.
Please refer to the highlights and comments in the marked-up manuscript copy.

Author Response
We thank you and the reviewers for appreciating our work. We have revised the manuscript per the comments and suggestions of the reviewers. We hope that the revised manuscript is suitable for publication in International Journal of Environmental Research and Public Health.
We have revised the manuscript according to the suggestions and comments. We have uploaded the Marked-up Manuscript (manuscript with revisions red highlighted), Manuscript (revised main text w/o track changes). Our responses to the comments are provided below in red.
My remaining concern is the copious repetition in text of content reported in tables. This must be avoided and I have provided some suggestions for revision. Additionally, conventional standards should be followed for reporting of statistical tests; again, I have provided an example of a possible revision.
- Response: We thank you for your comment. We revised everything as you recommended in the main text. Especially, we rewritten the descriptions of the statistical tests 3.1 and 3.2.
We have carefully considered the reviewers’ comments and have made the necessary revisions in the manuscript. We hope that the revised manuscript is suitable for publication in International Journal of Environmental Research and Public Health.
